# Implications of Dual Practice on Cataract Surgery Waiting Time and Rescheduling: The Case of Malaysia

**DOI:** 10.3390/healthcare9060653

**Published:** 2021-05-31

**Authors:** Weng Hong Fun, Ee Hong Tan, Sondi Sararaks, Shakirah Md. Sharif, Iqbal Ab Rahim, Suhana Jawahir, Vivien Han Ying Eow, Raoul Muhammad Yusof Sibert, Malindawati Mohd Fadzil, Siti Haniza Mahmud

**Affiliations:** 1Institute for Health Systems Research, National Institutes of Health, Ministry of Health Malaysia, Shah Alam 40170, Malaysia; jdreehong@moh.gov.my (E.H.T.); sararaks.s@moh.gov.my (S.S.); shakirah.ms@moh.gov.my (S.M.S.); fathullah@moh.gov.my (I.A.R.); suhana.j@moh.gov.my (S.J.); vivieneow@gmail.com (V.H.Y.E.); sitihaniza.m@gmail.com (S.H.M.); 2Hospital Pakar Sultanah Fatimah, Muar 84000, Malaysia; raoul@moh.gov.my; 3Medical Development Division, Ministry of Health Malaysia, Putrajaya 62590, Malaysia; drmalindawati@moh.gov.my

**Keywords:** dual practice, waiting time, out-of-office hours, cancellation, rescheduling, surgery, Malaysia

## Abstract

Background: Dual practice was implemented in selected Ministry of Health Malaysia hospitals to reduce brain drain and provide an alternative for patients willing to pay higher user fees to seek prompt treatment from the specialist of their choice. This study aimed to assess the implications of dual practice on waiting time and rescheduling for cataract surgery. Methods: A retrospective study was conducted in a referral hospital. Inpatient medical records of patients who underwent cataract procedures were used to study the waiting times to surgery and rescheduling between private and public groups. Results: Private patients had a considerably shorter waiting time for cataract surgery, seven times shorter compared to public patients where all surgeries were conducted after hours on weekdays or weekends. Additionally, 14.9% of public patients experienced surgery rescheduling, while all private patients had their surgeries as planned. The main reason for surgery rescheduling was the medical factor, primarily due to uncontrolled blood pressure and upper respiratory tract infection. Conclusion: Private service provision utilizing out-of-office hours slots for cataract surgery optimizes public hospital resources, allowing shorter waiting times and providing an alternative to meet healthcare needs.

## 1. Introduction

Malaysia adopts a two-tier public and private healthcare system [1,2] where the public sector is funded by general revenue and taxation, and where service is highly subsidized with nominal user charges. Inversely, the private sector is mainly driven by out-of-pocket payment and/or private health insurance [1]. Public sector healthcare providers are paid by salaries while their counterparts in the private sector are mainly remunerated through fee-for-service.

Similar to other countries, brain drain of healthcare professionals in Malaysia is common, especially among doctors and specialists [3]. Although strategies have been employed, including allowing full-time salaried doctors and specialists to practice in the private sector after working hours, the retention rate remains poor [1]. Private wings were established within public university hospitals, providing an avenue to serve patients with fee-for-service payments. Such a dual practice by healthcare providers is well known among healthcare providers across low-, and high-income countries [4,5,6].

Both negative and positive effects of the dual practice had been highlighted and debated, including access to healthcare service and quality of care, with conflicting evidence for the latter. The dual practice increased the quality of care in the public sector, as this encouraged skilled senior physicians’ retention [4,5,6]; however, the quality of care in terms of access and waiting times were compromised because physicians participating in dual practice paid more attention to their non-public patients [4,5,6].

For example, public hospitals in Australia provide both private and public healthcare services. Healthcare providers in the public healthcare sector benefit from monetary incentives by treating private patients. However, evidence showed that the treatment differed between private and public patients, whereby the former received higher priority, had longer intensive care unit stay hours and underwent more procedures despite being healthier and having fewer comorbidities [5]. In contrast, Wadee and Gilson [7] described the private ward system adopted in Tygerberg, South Africa, where there were no direct monetary benefits to the healthcare practitioner, as the extra revenue finances the hospital in the subsequent year. The tracer interventions found no fundamental differences between care provided to public and private patients [7].

Besides provider choice, concerns have been raised regarding crowded public hospitals with long waiting times for doctor visits and surgery [8]. Long waiting times for patients were a major health policy issue in many countries, including high-income countries such as the Organisation for Economic Co-operation and Development (OECD) countries [9]. Various solutions, such as redefinition of health personnel roles, refined working methods and restructuring of patient pathways, have been adopted to shorten waiting times [10].

Waiting time for surgery could be lengthened due to cancellation or rescheduling, hence affecting the quality of care, causing unnecessary stress to patients/caregivers, as well as possible harm to patients [11]. Moreover, this can result in a waste of resources [12,13], which is one of the parameters used to assess the quality of care and healthcare management [14,15].

A review reported that cancellation rates could range from 0.48% to 38% [16], and surgery cancellation varied across countries; such as 4.4% in Lebanon [17], 7.6% in Saudi Arabia [18], 17.6% in India [19], 17.5% in China [20] and 21.9% in Burkina Faso, in the sub-Saharan Africa region [21]. Common reasons reported for surgery cancellation include insufficient pre-surgery preparation and assessment, patient factors (e.g., patient unfit, non-compliance of pre-operative instructions) and facility-related issues (e.g., lack of operation theatre time, equipment issues), as well as healthcare provider-related issues (e.g., unavailability of surgeon/anesthetist) [17,18,20,22].

To address these issues, Ministry of Health (MOH) Malaysia initiated a dual practice program within the public sector in 2007 [23]. This initiative aimed to address brain drain and reduce crowding as well as waiting times in public service. It allows qualified, salaried senior specialists to treat private patients in the MOH facilities after working hours under strict regulations and close monitoring of implementation. For outpatient consultations, private patients are seen after public patients, with the number not exceeding 30% of total patients. Surgical procedures for private patients could be conducted after working hours. Apart from their monthly salaries, specialists providing private service in MOH hospitals are remunerated via a fee-for-service [23].

Private patients pay substantially high user fees in exchange for the freedom to choose their attending specialists and quicker access to healthcare services, and enjoy first-class or superior facilities, the latter subject to resource availability [23]. However, this has raised concerns about the possible inequalities in healthcare service provision in public hospitals [3]. With limited knowledge on the effect of the dual practice system to healthcare services in an upper middle-income country like Malaysia, this study aimed to assess the implications of dual practice on waiting times and rescheduling for cataract surgery in a public hospital.

## 2. Materials and Methods

### 2.1. Study Design and Setting

We reviewed electronic medical records in one of the pioneer hospitals with dual practice. The study hospital is a paperless hospital, supported by Total Hospital Information System (THIS), an integration of clinical, administrative and financial systems. Among the thirteen departments with private inpatients, cataract cases were assessed because of the relatively high private patient load and the involvement of elective procedures suitable for waiting time to surgery comparison.

### 2.2. Outcome Measures

We used two outcome measures to assess gaps in service provision for public and private groups: waiting time for elective surgery and rescheduling of surgery.

#### 2.2.1. Waiting Time for Elective Surgery

We measured waiting times in three ways:Clock-continuous scenario: continuous measurement of waiting times in all patients regardless of their reasons for rescheduling [9].Clock-pause scenario: discounts the rescheduling period in patients rescheduled due to patient or medical factors [24].Clock-restart (“new clock”) scenario: eliminates the previous waiting time and restarts a new clock when there is rescheduling [25].In these three definitions, the starting point was patient enlistment into the procedure list, and the end time was the date the procedure was done. The illustration of these definitions is depicted in Figure 1, adapted from Viberg et al., 2013 [26].

#### 2.2.2. Rescheduling of Surgery

This study considered elective surgery as rescheduled when a patient had received confirmation of the surgery’s date but the surgery was subsequently rescheduled and performed at a later date (adapted from the NHS Modernisation Agency Theatre Programme, 2002 [27]). We counted the episode(s) of surgery rescheduling and collected reasons for the rescheduling, and the latter was classified into medical, resource, or patient factors. Medical factors were defined as clinical conditions in which the patient was not ready for surgery [24] or a clinical decision not to treat was made [25], whilst patient factors include personal (non-clinical) patient reasons, e.g., work commitment [24]. Resource factors were due to hospital-related issues, e.g., operating theatre time, staffing, equipment, scheduling error or bed availability [17,24].

### 2.3. Sample Size Calculation

There was limited evidence on the average elective cataract surgery waiting time in Malaysia, thus international data were used to estimate the sample size. To determine the desired precision for the mean waiting time, we used the formula [28] *N* = (Zσ/E)^2^, where Z = 95%, being the confidence level, standard deviation (σ) of 9.9 days [29], and the desired margin of error (E) of 3 days, the estimated minimum sample size was 42 cases.

As compared with public patients, the number of private patients extracted from the financial system (provided by the Financial Department) were relatively small; therefore, we included all private patients in 2016 who fulfilled the inclusion criteria. Random selection was performed for public groups with a 3-fold increase to accommodate for the missing data, patient dropout, misclassification and variability of rescheduling. Random selection was done using the Statistical Package for the Social Science software version 23 (SPSS v23, IBM, Armonk, NY, USA).

### 2.4. Data Collection

Data of public and private inpatients scheduled for cataract procedures in 2016 were extracted from the administrative system (provided by the Medical Record Department) in June–July 2017. From the administrative system, the variables extracted for cross-checking with the financial system included patient registration number (RN) for linking to clinical information, type of appointment, admission date, discharge date, type of diagnosis, ward and bed class. From the variable of type of diagnosis, inpatients’ diagnosis, according to the International Statistical Classification of Diseases and Related Health Problems (ICD) [30], ICD10 H26–28 was extracted, while the variable for type of appointment enabled us to identify private or public patients. Using the RN for the selected patients, clinical information was extracted from the electronic medical records. This consisted of patient information such as diagnosis (a string variable, used to verify the ICD classification), co-morbidity, type of surgery, date confirmed for surgery and date of surgery. Additionally, details on the frequency and reasons for rescheduling of surgery were obtained. All personal identifiers were de-identified after data verification.

### 2.5. Data Analysis

Descriptive and inferential statistics (Chi-square test, Mann–Whitney U test and Fisher’s Exact Test) were conducted to examine the waiting time and surgery rescheduling for public and private groups, using the SPSS v23. We excluded patients with missing or misclassified public or private status, patients yet to undergo surgery as of May 2017 and patients with unknown rescheduling reasons.

## 3. Results

### 3.1. Socio-Demographic Characteristics

Of the 323 cataract patients who received treatment during the study period, 270 and 53 were public and private groups, respectively. For the private group, all cases in 2016 who fulfilled the inclusion criteria were included for analysis. For the public group, twelve cases were excluded from waiting time analysis (Figure 2).

Both private and public patients for cataract procedures were in their 50s, reflecting the nature of the disease. Only a small number of the private service users (*n* = 5) were non-Malaysians. Ethnicity profile for the private group was different from that of public group, with less Malay patient uptake in the private service. A greater percentage of comorbidities was also seen among the public patients (Table 1).

### 3.2. Waiting Time to Operation

Table 2 shows the significant difference between the public and private waiting times for all three different definitions. Using the clock-continuous approach for cataract surgery, the median waiting time for private groups was about seven times shorter than for public groups.

Using the clock-pause approach, the gap between public and private waiting times was reduced. Waiting time for cataract surgery was six times shorter for the private group. If the clock-restart approach was used, the ratio was even less than the clock-pause approach.

### 3.3. Surgery Rescheduling

Among the public cases, 14.9% were rescheduled, while there were no private cases with rescheduling (Table 3). The main reason for rescheduling was medical factors (76.0%), primarily attributed by uncontrolled blood pressure, whereas equipment issues were the reasons for resource factor (Table 4).

## 4. Discussion

Our study showed that private patients experienced a shorter waiting time compared to public patients with the use of out-of-office hours slots and were not affected by surgery rescheduling.

Waiting time, the main quality indicator of healthcare service delivery, is routinely used to study patient access to healthcare [31,32]. The Department of Health, United Kingdom (UK), sets an 18-week referral-to-treatment time target waiting time from General Practitioner (GP) referral to treatment in hospital, to be achieved by all state hospitals as a measure of good practice [33]. Meanwhile, the waiting time target of the OECD countries ranges from 30 days to 6 months [9,34]. Due to different methods used in measuring elective surgery waiting time, comparisons across countries are difficult. Using clock-continuous and clock-pause approaches, comparison with international literature suggests the median waiting time for public cataract surgery in this study was relatively shorter than some high-income countries. With the clock-continuous definition, the public patient waiting time for cataract surgery was 48 days, which was faster than England (59 days), Scotland (62 days), Australia (91 days) and Spain (89 days) [9]. The clock-pause approach showed the waiting time for cataract surgery to be 41.5 days for public patients, compared to Canada (49 days), New Zealand (88 days) and Finland (111 days) [9].

A substantially shorter waiting time was seen among the private patients, similar to Australia, where those under private health insurance in public hospital had a shorter waiting time (median 30 days) than public patients (median 109 days) [35]. Some countries have established national waiting time guarantees for elective surgery. For example, waiting time guarantees for cataract surgery in Spain and Canada are 180 and 112 days, respectively [36,37]. In Malaysia, waiting time for cataract surgery was set as a key performance indicator for ophthalmology services, with the target of >80% of patients expected to have appointments given for cataract surgery within 16 weeks [38]. Our study showed that the performance of the study hospital far exceeded the national waiting time target.

Procedure rescheduling prolongs waiting time and causes unnecessary anxiety and emotional distress in patients, as well as to their caregivers and families [39,40]. In this study, medical conditions were found to be the main contributing factor for rescheduling. In contrast, a study by Ezike et al. [41] reported that physician-related factors, such as surgeons’ unavailability, contributes to the highest incidence of elective surgery cancellation in a Nigerian hospital, followed by patient-related, administrative/logistics and medical factors. More importantly, there is a need to know whether rescheduling of surgeries was avoidable; and whether pre-operative diagnostic assessment and coordination between all healthcare providers/departments involved were well-planned to avoid rescheduling [39,40]. These issues, though pertinent, were not investigated in our study.

Implications of the dual practice on access, quality, efficiency and cost of services have been studied [5,6]. It has been argued that health care systems with a fixed remuneration to providers was more prone to suffer queues compared to the systems using fee for service [36]. The existence of waiting lists is due to the presence of demand and supply mismatch, i.e., the variation in supply management in healthcare systems could lead to long waiting times [42]. Internationally, the public–private mix in Ireland has led to the development of two separate waiting lists for public and private patients, where private patients had comparatively shorter waiting times [43]. Similarly, in Australia, public patients had to wait longer for elective surgery as compared to private patients in public hospitals [35], whilst the two-tier charging system in Zambian public hospitals brought about a preferential allocation of resources to patients paying higher fees [44]. With the private service provision within the public section, it is expected to decrease the public patient queue by creating an alternative waiting list while optimizing human and physical resources with the out-of-usual hour policy [23].

In this study, private patients experienced substantially shorter waiting times and no rescheduling. At what cost does this improved performance come with, in terms of equity and economic cost? Did we follow Zambia’s footstep creating a disproportionate allocation of resources, favoring those able to pay? [44] The UK health system reforms, including private service provision, have improved the allocation efficiency and driven down long waits of the poor [45]. In New Zealand, the opposite effect was shown, where high private provision of elective surgeries did not improve access to publicly funded services for the poor [46]. In essence, the private service may result in inequitable treatment as the provision of healthcare services depends on the ability to pay instead of on clinical needs.

It was also highlighted that the dual practice may foster shirking duties in the public sector and inappropriate use of public resources [4,5,6]. The implementation of private service in Malaysia is strictly regulated and monitored, with clear guidelines on limiting the schedule, rationing the number of patients and restricting the earning from the provision of the private service to ensure that the public interest is protected and remains the main focus [23]. The services offer more options for private patients in terms of choice of experienced specialists and faster access to care; along with additional income for specialists [5,6]. Additionally, the unsubsidized private user fees in public hospitals are still at a competitive price than the user fees in private hospitals [23,47,48]. Such an initiative might not provide direct benefits to public patients; however, policies such as reimbursement for conducting elective surgeries on Saturday for public patients [49] could improve the waiting time for public patients. In terms of “brain drain” within the public sector, whether the private provision within the public sector could retain experienced providers was still under-explored and its positive impacts were still questionable [50].

As for the government, this not only lessens the government’s burden in subsidizing healthcare, but also generates income for the government by recovering a proportion of the revenues from private services. No doubt, the increased utilization of out-of-office hour facilities (operation theatre, equipment and utilities) and human resources (specialist service and surgical team) could possibly bring about a positive impact on the economy. Yet, this also implicates the fundamental philosophy of public service provision and the financial impact with out-of-hours physical and human resource utilization. Our study did not compare the waiting times before and after the implementation of the private service within public provision. Further research is needed to explore the impacts on waiting times for public and private patients in various dimensions.

This study used electronic medical records, a method both convenient and inexpensive, to answer time-sensitive questions. Data coding for patient databases remained unchanged during the study period and all inpatients were ICD-coded upon discharge, hence limiting the likelihood of data miscoding. Data verification for included public/private patient status was done using three data sources: administrative database, medical records and financial records. In addition, a clear temporal relationship between disease and surgery can be demonstrated by a retrospective review of electronic medical records. Such single center studies limit results’ generalizability across other centers providing a similar service. Data inconsistencies and missing data were observed as they were collected from medical record database retrospectively. The identification of the point of primary care referral and/or the earliest point requiring intervention was not possible owing to the lack of data linking to primary care facilities, hence affecting the waiting time definition. The reasons for rescheduling were solely based on medical records. We did not validate the reasons documented with the patients or caregivers. There may be a lack of documentation for rescheduling. These situations warrant further investigation to reduce avoidable rescheduling.

## 5. Conclusions

Optimizing out-of-office hour use of public hospital resources for cataract surgery provision could contribute to shorter waiting times. As long as waiting times for elective surgery have always been a public concern, managing and balancing the provision of the public and private services is crucial to facilitate timely access to care.

## Figures and Tables

**Figure 1 healthcare-09-00653-f001:**
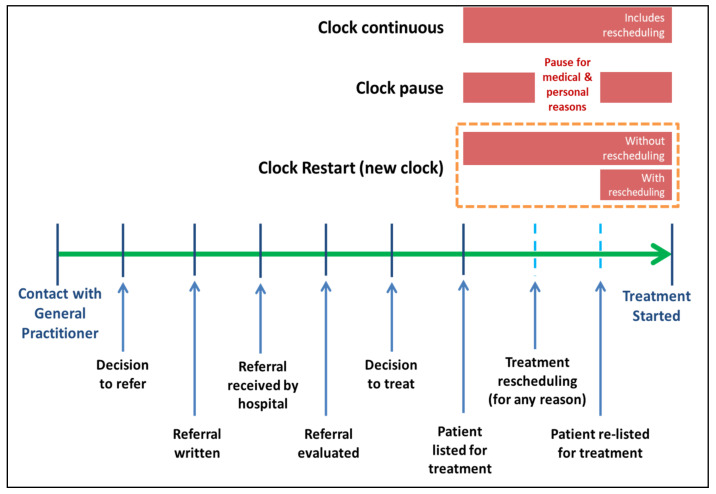
Waiting time definitions for clock continuous, clock pause and clock restart. Adapted from Viberg et al., 2013 [26].

**Figure 2 healthcare-09-00653-f002:**
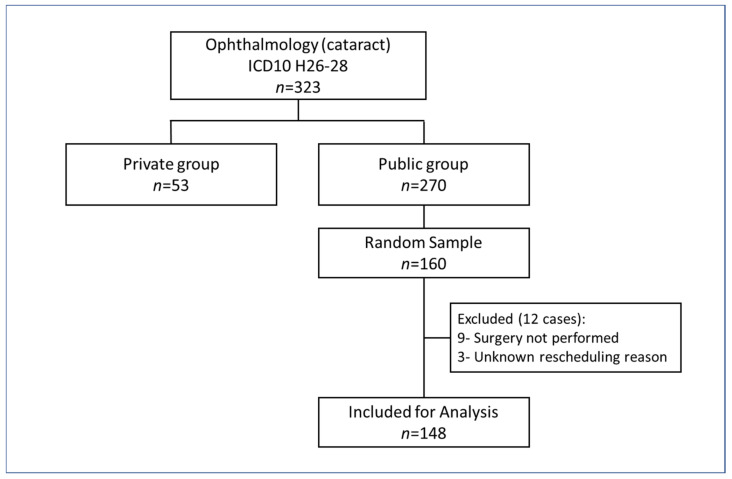
Flow chart of sampling method for cataract patients.

**Table 1 healthcare-09-00653-t001:** Socio-demographic characteristics of public and private groups.

Characteristic	Cataract (ICD10 H26–28)		*p*-Value
	Private	Public	
Age (year) [Mean (SD)]	58.66 (14.41)	50.44 (22.56)	0.166 ^a^
Gender (%)			0.625 ^b^
Male	30 (56.6)	78 (52.7)	
Female	23 (43.4)	70 (47.3)	
Nationality (%)			0.001 *^,c^
Malaysian	48 (90.6)	148 (100)	
Non-Malaysian	5 (9.4)	0 (0)	
Ethnicity (%)			<0.001 *^,c^
Malay	13 (24.5)	80 (54.1)	
Chinese	17 (32.1)	41 (27.7)	
Indian	18 (34.0)	24 (16.2)	
Others ^d^	5 (9.4)	3 (2.0)	
Comorbidities ^e^ (%)			0.015 *^,b^
Yes	32 (60.4)	115 (77.7)	
No	21 (39.6)	33 (22.3)	
Surgery Performed Time (%)			<0.001 *^,c^
Office hour	0 (0)	117 (79.0)	
After hours on weekdays/weekends	53 (100.0)	29 (19.6)	
Unknown	0	2 (1.4)	

Note: * *p* < 0.05, ^a^ Mann–Whitney U test, ^b^ Chi-square Test, ^c^ Fisher’s Exact Test, ^d^ Category for Others includes 5 non-Malaysians in the private group and 3 Malaysians of other ethnicities in the public group. ^e^ Presence of one or more conditions/disorders co-occurring with cataract such as hypertension, diabetes, hyperlipidaemia, cardiovascular diseases, hyperthyroidism, asthma and renal diseases.

**Table 2 healthcare-09-00653-t002:** Median and mean waiting time to cataract procedure (in days) for public and private groups.

Waiting Time to Cataract Procedure Approaches (Days)		Private (*n* = 53)	Public (*n* = 148)	Public: Private Ratio ^a^
Clock continuous	Median (Q1, Q3)	7 (2.00, 14.00)	48.00 (16.00, 94.00)	6.9 *
	Mean (SD)	15.26 (27.28)	70.57 (82.50)	
Clock pause	Median (Q1, Q3)	7 (2.00, 14.00)	41.5 (15.50, 89.50)	5.9 *
	Mean (SD)	15.26 (27.28)	60.48 (59.69)	
Clock restart (new clock)	Median (Q1, Q3)	7 (2.00, 14.00)	38.5 (13.00, 83.00)	5.5 *
	Mean (SD)	15.26 (27.28)	57.41 (66.14)	

Note: * *p*-value < 0.001, ^a^ Mann–Whitney U test was performed to compare the waiting time to procedure.

**Table 3 healthcare-09-00653-t003:** Rate and reasons for cataract surgery rescheduling.

	Private	Public
Scheduling status ^a,^*(%)		
Scheduled	53 (100)	126 (85.1)
Rescheduled	0	22 (14.9)
Number and reason for rescheduling (%)		
One time	0 (0)	19 (86.4)
*Medical*		*14 (73.7)*
*Patient*		*3 (15.8)*
*Resource*		*2 (10.5)*
Two times	0 (0)	3 (13.6)
*Medical*		*2 (66.7)*
*Medical and Patient*		*1 (33.3)*

Note: * *p*-value < 0.001, ^a^ Chi-square Test.

**Table 4 healthcare-09-00653-t004:** Reasons for cataract surgery rescheduling for public group.

Reason *	Count	%
Medical Factor		
Uncontrolled blood pressure	8	38.1
Upper Respiratory Tract Infection	5	23.8
Cardiac condition	1	4.8
Uncontrolled diabetes	3	14.3
Other medical factors	3	14.3
Additional investigation required	1	4.8
Total	21	100.0
Patient Factor		
Patient’s request or due to other personal reasons	4	100.0
Total	4	100.0
Resource Factor		
Operation theatre chiller not functioning	1	50.0
To try new phacoemulsification machine	1	50.0
Total	2	100.0

* A rescheduling event could have more than one reason.

## Data Availability

Data that support the findings of this study are available from the Ministry of Health Malaysia, but restrictions applied to the availability of these data, which were used under license for the current study, and so are not publicly available. The data are, however, available from the authors upon reasonable request and with the permission of the Ministry of Health Malaysia.

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
