# Peer review of "Implications of Dual Practice on Cataract Surgery Waiting Time and Rescheduling: The Case of Malaysia"

_healthcare, 2021, doi:10.3390/healthcare9060653_

Round 1

Reviewer 1 Report

The authors have presented a research about the implications of dual practice on service provision for cataract surgery. There are concerns that need to be addressed by the authors in order to improve the quality of the manuscript.

1. There are many errors of English and grammar. An English editorial service may be solicited to address this issue. 

2. Abstract: In line 16, the phrase "public provision" is confusing. Revise. Also, in line 20, revise "procedures was used" to "procedures were used".

3. The study aim as stated in the abstract is ambiguous and not consistent with that presented in the main text. The study aim should be precise and "measurable" for statistical analyses. Revise the study aim. The terms in the manuscript are not measurable:

"to assess the implications of dual practice on public provision for surgery and rescheduling occurrence, between public patients paying nominal user fees and private patients paying higher user fees in the hospital".

" this study aimed to assess the degree of disparities of dual practice within public provision, in terms of waiting time to surgery and rescheduling.

4. The independent variables are not clearly defined in the paper. How did you determine "private" and "public" patients from the medical records?

5. Statistical analyses: This is an observational study that is not testing any hypothesis or determining the number of participants needed to detect a  relevant  effect. Why did the authors calculate the sample size?

How are we sure that there are statistically significant differences in waiting times and rescheduling between the two groups when there were no tests done and significant levels stated? No t-test, or ANOVA  was conducted.

6. Little or nothing is stated in the discussion to address the disparities caused by dual practice on service provision for cataract surgery. What can be done to assist public patients benefit from the dual practice?

Best of luck!

Author Response

Dear Reviewer,

Re: Manuscript Healthcare-1196748

Thank you for the opportunity to revise and resubmit our manuscript titled “Implications of dual practice on service provision for cataract surgery”. We have provided a point-by-point response to the reviewers’ comments. Please refer to the file attached. 

Yours sincerely,

Dr Weng Hong Fun, on behalf of all authors

Reviewer 2 Report

The authors have conducted a descriptive study comparing the waiting time of public and private cataract patients in Malaysia. This study has an important implication for countries implementing dual practice system to avoid brain drain like Malaysia. The manuscript is well-written overall. I only have several minor comments which would hopefully be helpful to improve the manuscript.

  • Title: “Service provision” is a very broad term. Using this term in the title might be a little misleading to the readers as the main outcome variable in this study is waiting time. I would suggest the authors to reword the title a little.
  • Lines 45, 97: In global health and public health manuscripts, I would strongly suggest the authors to avoid using the terms “developing”, and “developed”. Instead, use the income level defined by the World Bank when describing the economic status of a country. Malaysia is, I believe, categorized as an upper-middle income country.
  • I am still not clear of the sample size calculation. What is this sample size calculation for? I suggest the authors to be more detailed in the descriptions of the sample size calculation. To detect the difference in waiting time of private and public patients? To identify the proportions? Descriptions like these are helpful.
  • Data analysis: I would suggest the authors use the term “descriptive statistics” instead of “descriptive analysis”. In addition, the authors only computed the percentages of private and public patients and no statistical tests were performed. This would see to me like a general report and not a research paper. I would recommend the authors perform some statistical tests.

Author Response

Dear Assistant Editor/ Reviewers,

Re: Manuscript Healthcare-1196748

Thank you for the opportunity to revise and resubmit our manuscript titled “Implications of dual practice on service provision for cataract surgery”. We have provided a point-by-point response to the reviewers’ comments. Please refer to the file attached.  

Yours sincerely,

Dr Weng Hong Fun, on behalf of all authors

Round 2

Reviewer 1 Report

The manuscript has been substantially improved by the revisions.

However, in line  23, include the percentage of the private group that had surgery as planned. In line 66, correct "...long waiting times has..." to "...long waiting times have..." It is still not clear what variables in the medical record were used to identify public from private patients...Please clarify. In line 169, state the exact inferential statistics that was done.

Best of luck!

Author Response

Date: 6 May 2021

Reviewer 1

MDPI Healthcare Editorial Office

St. Alban-Anlage 66, 4052 Basel, Switzerland

Dear Reviewer 1,

Re: Manuscript Healthcare-1196748

Thank you for the opportunity to revise and resubmit our manuscript titled “Implications of dual practice on service provision for cataract surgery”. We have provided a point-by-point response to the reviewer’ comments. Kindly refer to the attachment.

Yours sincerely,

Dr Weng Hong Fun, on behalf of all authors
